# Breakthrough photothermal ammonia decomposition via low-barrier Ni-CeO$_{2-x}$ interfaces on carbon nanotubes

Ruike Tan[1,2], Xiaowei Mu [1,2] ✉, Xinhui Wang[1,2], Yuxiang Kong[1,2], Qing Ji[1,2], Qingyun Zhan[1,2], Qingchuan Xiong[1,2] & Lu Li [1,2] ✉

Efficient ammonia decomposition is crucial for hydrogen economy, but inexpensive Ni catalysts require excessively high temperatures due to limited N-N coupling. Here, we overcome this challenge by constructing a photothermal catalyst with closely interfaced, defect-rich CeO$_{2-x}$ nanodomains and electron-rich Ni nanoparticles on carbon nanotubes. The Ni-CeO$_{2-x}$/CNTs catalyst achieves a hydrogen production rate of 298.4 mmol g$_{cat}^{-1}$ min$^{-1}$ under full-spectrum light irradiation, which exceeds that of most reported Ru catalysts, and maintained stable activity for over 50 h in continuous-flow operation. The high performance arises from the synergistic effect of thermally promoted photocatalytic N-H bond cleavage and a largely reduced N-N coupling barrier, enabled by efficient photothermal conversion of carbon nanotubes and the up-shifted d-band center of the Ce-Ov-Ni interface (Ov = oxygen vacancy).

The hydrogen economy is hindered by challenges in storage and transport, primarily due to hydrogen's low volumetric energy density and safety concerns associated with its explosiveness[1]. Ammonia (NH$_3$) has emerged as a promising hydrogen carrier, offering a high hydrogen content (17.8 wt.%), facile liquefaction under mild conditions (8–10 bar at 20 °C), and a carbon-free decomposition route (2NH$_3$ → N$_2$ + 3H$_2$)[2]. Nickel-based catalysts, as cost-effective non-noble metal alternatives, are attractive for catalytic ammonia decomposition. However, their performance is limited by the high energy barrier associated with N-adatom recombination and desorption, necessitating elevated reaction temperatures and strong resistance to sintering[3].

Light-driven catalysis offers a sustainable strategy for overcoming activation barriers by harnessing photon energy, enabling high reactivity under mild conditions[4–6]. In photocatalytic ammonia decomposition, light primarily excites the photocatalyst to generate electron−hole pairs, where photogenerated holes serve as strong oxidants that effectively promote N−H bond cleavage[7,8]. However, another key step−N-adatom recombination and desorption−requires substantial thermal energy to overcome its high activation barrier[9–12]. Purely photocatalytic systems lack sufficient heat to drive this step

efficiently, resulting in sluggish reaction kinetics and extremely low hydrogen evolution rates (<0.01 mmol g$_{cat}^{-1}$ min$^{-1}$)[7,13–17]. In contrast, photothermal catalysis, through the synergistic interplay of light and heat, effectively addresses this limitation[18–20]. In such systems, light energy facilitates N−H bond activation, while thermal energy promotes N−N coupling, collectively enabling efficient and low-temperature ammonia decomposition with significantly enhanced reaction rates.

An ideal photothermal catalyst for ammonia decomposition should integrate several key attributes to fully exploit the synergistic effect of light and heat[21]. First, it must exhibit superior light-harvesting capability and efficient charge separation to convert ultraviolet and visible light into chemical energy with high efficacy. Second, the catalyst should feature rationally designed active sites that facilitate ammonia adsorption, promote N−H bond cleavage, and optimize the N−N coupling to lower activation barriers and enhance the efficiency of the overall reaction pathway. Additionally, it should efficiently convert low-energy light−particularly near-infrared (NIR), which cannot drive charge separation−into heat and deliver it to active sites, transforming otherwise wasted photons into a driving force for N−N coupling and enhanced charge carrier dynamics. Long-term resistance to ammonia and light-induced corrosion is also essential to ensure structural and

[1]College of Chemistry, Jilin University, Changchun, People's Republic of China. [2]State Key Laboratory of Inorganic Synthesis and Preparative Chemistry, College of Chemistry, Jilin University, Changchun, People's Republic of China. ✉e-mail: lilymu125@163.com; luli@jlu.edu.cn

functional stability. Moreover, the use of earth-abundant transition metals such as Ni or Fe is vital for scalable and sustainable applications. Realizing all these features in a single catalyst, however, remains a significant challenge.

Herein, we develop one of the most efficient photothermal catalysts to date, achieved by integrating $CeO_{2-x}$ nanodomains and Ni nanoparticles onto CNTs with close interfacial contact. The exceptional performance originates from the synergistic combination of UV-visible photocatalysis driven by $CeO_{2-x}$ and rationally engineered interface sites. CNT's remarkable light-to-heat conversion and electron transport capabilities significantly enhance the photoexcited carrier separation in $CeO_{2-x}$, thereby accelerating the photocatalytic N−H bond cleavage. The formed $Ce−O_V−Ni$ interface upshifts the Ni d-band center, promoting the adsorption of N intermediates and significantly reducing the N−N coupling barrier from 1.34 eV (in conventional Ni catalysts) to 0.53 eV, completely breaking the intrinsic limitation of Ni-based catalysts.

## Results

### Structures and properties of catalysts

A series of $CeO_2$ modified carbon nanotube (CNTs) composites were synthesized via a liquid-phase alkaline precipitation method (Fig. 1a). The target composite, denoted as $CeO_{2-x}$/CNTs, was obtained by adjusting the mass ratio of $CeO_2$ to CNTs to 0.5 ($m_{CeO2}/m_{CNTs} = 0.5$), while pure $CeO_2$ and CNTs were served as control samples. Subsequently, 10 wt.% Ni was deposited onto the composites via a $NaBH_4$-assisted chemical reduction method, yielding the Ni-$CeO_{2-x}$/CNTs, Ni/$CeO_2$, and Ni/CNTs catalysts, respectively. Inductively coupled plasma optical emission spectroscopy (ICP-OES) analysis confirmed that the actual Ni loading of all catalysts was in close agreement with the nominal values (Supplementary Table 1).

Powder X-ray diffraction (XRD) patterns (Supplementary Fig. 1) reveal the coexistence of characteristic peaks corresponding to the cubic fluorite structure of $CeO_2$ and the graphitic features of CNTs in $CeO_{2-x}$/CNTs[22]. Compared to pure $CeO_2$, the diffraction peaks of $CeO_2$

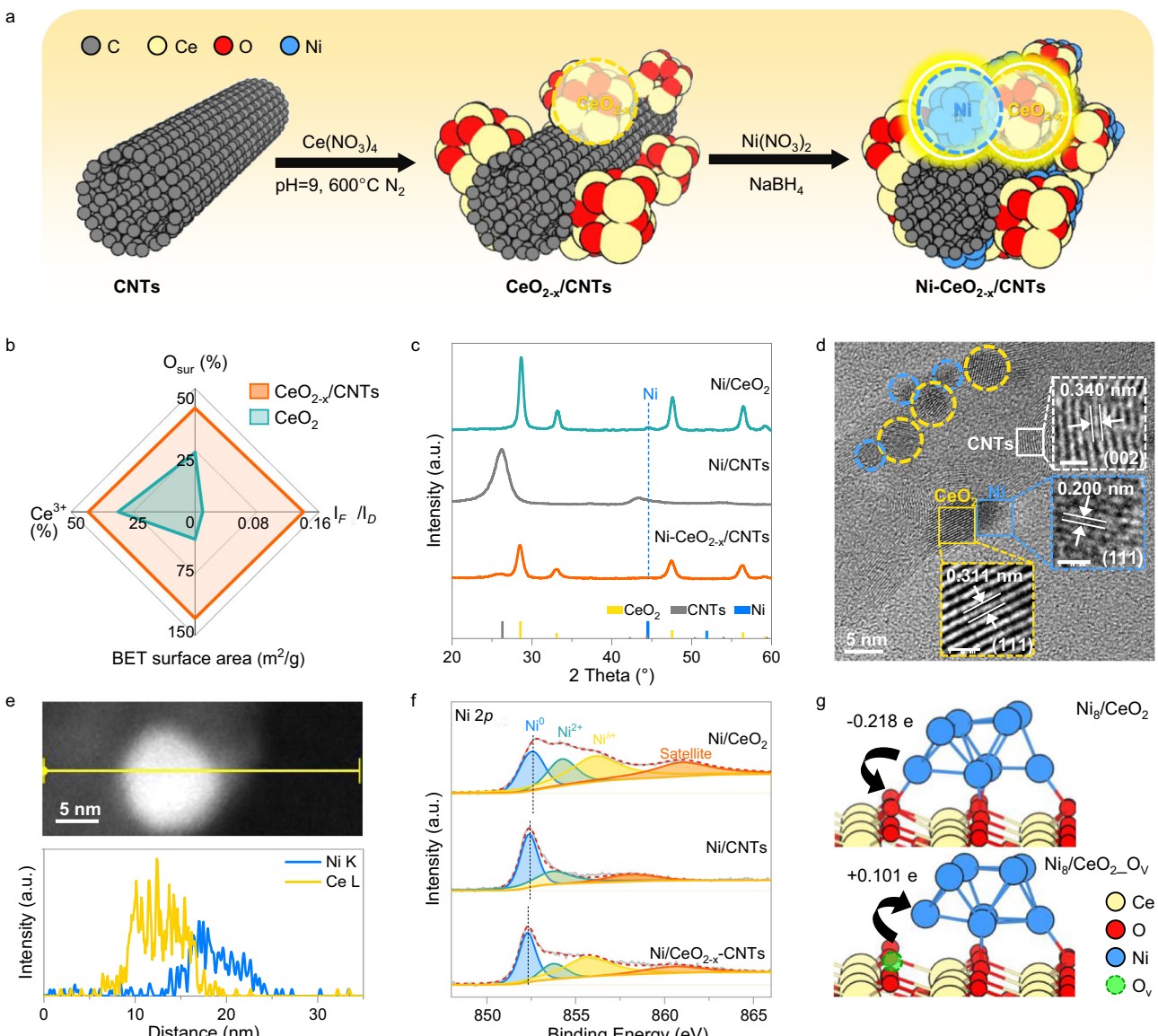

**Fig. 1 | Structures and electronic characterization of catalysts. a** Schematic fabrication process of Ni-$CeO_{2-x}$/CNTs. **b** Comparative properties of $CeO_{2-x}$/CNTs versus $CeO_2$. The detailed values are summarized in Supplementary Table 2. **c** Powder XRD patterns of Ni/$CeO_2$, Ni/CNTs, and Ni-$CeO_{2-x}$/CNTs. **d** HRTEM images of Ni-$CeO_{2-x}$/CNTs. **e** HAADF-STEM image and corresponding elemental line scanning of Ni-$CeO_{2-x}$/CNTs. **f** Ni $2p_{3/2}$ XPS of Ni/$CeO_2$, Ni/CNTs and Ni-$CeO_{2-x}$/CNTs. **g** Bader charge(|e|) analysis between $Ni_8$ cluster and $CeO_2$ surface without/with (up/down) oxygen vacancy. C: gray, Ce: yellow, O: red, Ni: blue, and $O_v$: green spheres.

in CeO$_{2-x}$/CNTs are significantly broadened, indicating a reduced crystallite size. Transmission electron microscopy (TEM) images (Supplementary Fig. 2) further confirm that CeO$_{2-x}$ is uniformly dispersed on the CNTs surface with an average particle size of 6.6 nm–substantially smaller than that of pure CeO$_2$ (17.4 nm). This size reduction can be attributed to the high surface area of CNTs (Supplementary Fig. 3), which provides abundant anchoring sites and suppresses CeO$_2$ crystal growth[23–25]. Raman spectroscopy (Supplementary Fig. 4) reveals a pronounced defect band (D-band) at 620 cm$^{-1}$ in CeO$_{2-x}$/CNTs, with an intensity ratio (I$_D$/I$_{F2g}$) of 0.14 relative to the intrinsic $F_{2g}$ mode of CeO$_2$ at 467 cm$^{-1}$–14 times higher than that of pure CeO$_2$. Quantitative analysis of oxygen vacancies was performed by X-ray photoelectron spectroscopy (XPS) of the O 1$s$ and Ce 3$d$ regions (Supplementary Fig. 5). The O 1$s$ spectrum indicates that oxygen vacancies account for 42% of the total oxygen species, while the Ce$^{3+}$/(Ce$^{3+}$ + Ce$^{4+}$) ratio derived from the Ce 3$d$ spectrum reaches 43%. These results, summarized in Fig. 1b and Supplementary Table 2, demonstrate the formation of highly dispersed, defect-rich CeO$_{2-x}$ nanodomains on the CNTs surface.

The nature of Ni nanoparticles was elucidated by combining XRD patterns, TEM images, and XPS spectra. As shown in the XRD patterns (Fig. 1c), distinct metallic Ni diffraction peaks are observed only for Ni/CeO$_2$, indicating that Ni species in Ni/CNTs and Ni-CeO$_{2-x}$/CNTs exhibit smaller particle sizes and higher dispersion. TEM images (Supplementary Fig. 6) confirm that the average Ni particle sizes on CeO$_2$, CNTs, and CeO$_{2-x}$/CNTs are 10.2 nm, 5.8 nm, and 5.5 nm, respectively. Notably, high-resolution TEM (HRTEM, Fig. 1d) and elemental line-scanning images (Fig. 1e) clearly reveal that in Ni-CeO$_{2-x}$/CNTs, Ni nanoparticles (highlighted in blue) preferentially anchor around CeO$_{2-x}$ nanodomains (highlighted in yellow) on the CNT surface, forming intimate Ni-CeO$_{2-x}$ interfaces. This phenomenon can be attributed to strong metal-support interaction (SMSI) between Ni and CeO$_2$, which stabilizes Ni nanoparticles through interfacial Ce−O−Ni bonding[26–28]. XPS analysis of the Ni 2$p_{3/2}$ region (Fig. 1f) indicates a higher proportion of Ni$^{2+}$ and Ni$^{\delta+}$ species in Ni-CeO$_{2-x}$/CNTs compared to Ni/CNTs, further supporting the formation of the Ni-CeO$_{2-x}$ interface[29].

Ultraviolet photoelectron spectroscopy (UPS, Supplementary Fig. 7) shows a decreasing trend in work function (WF) across the support materials: CeO$_2$ (5.79 eV), CNTs (4.91 eV), and CeO$_{2-x}$/CNTs (4.71 eV). Correspondingly, Ni-CeO$_{2-x}$/CNTs exhibits a downward shift in Ni 2p binding energy (Fig. 1F), showing 0.25 eV and 0.07 eV lower values compared to Ni/CeO$_2$ and Ni/CNTs, respectively. Density functional theory (DFT) calculations of work function (WF$_{cal.}$, Supplementary Fig. 8) further support this trend, indicating that oxygen vacancy-rich CeO$_2$_O$_V$ (4.63 eV) and CNTs (4.72 eV) within the CeO$_{2-x}$/CNTs composite synergistically act as electron donors, promoting electron transfer to metallic Ni (5.17 eV). In contrast, pure CeO$_2$ exhibits a higher work function (5.94 eV) and behaves as an electron acceptor. Bader charge analysis and differential charge density maps (Fig. 1g and Supplementary Figs. 9 and 10) further corroborate these findings, showing that CeO$_2$_O$_V$ donates 0.101 |e| to Ni$_8$, while pure CeO$_2$ withdraws 0.218 |e| from Ni$_8$. These experimental and theoretical results collectively confirm the enhanced electron-donating capability of CeO$_{2-x}$/CNTs-supported Ni nanoparticles.

Finally, we constructed a well-defined Ni-CeO$_{2-x}$/CNTs catalyst with closely interfaced, defect-rich CeO$_{2-x}$ nanodomains and highly dispersed Ni nanoparticles on carbon nanotubes (CNTs). This architecture combines spatial confinement from CNTs, strong Ce−O−Ni interfacial interactions for sintering resistance, and directional electron transfer that enriches Ni active sites.

## Photothermal catalytic ammonia decomposition

Photothermal ammonia decomposition was initially conducted in a batch reactor under visible-IR irradiation (300 W xenon lamp equipped with a 400 nm cut-off filter, 1.4 W cm$^{-2}$) without external heating (Supplementary Fig. 11). As shown in Supplementary Fig. 12, Ni-CeO$_{2-x}$/CNTs exhibited the highest decomposition efficiency among a broad range of photothermal catalysts with varied supports, oxides, and metal centers. Comparative studies revealed that its hydrogen production rate significantly surpassed those of single-component catalysts (Ni-CeO$_2$ and Ni/CNTs) and their physical mixture (Ni-CeO$_2$ + Ni/CNTs), underscoring the importance of the synergistically integrated nanocomposite architecture in boosting photothermal activity (Fig. 2a). No hydrogen was detected over metal-free CeO$_{2-x}$/CNTs, confirming that metal active sites are essential for the reaction. Upon light irradiation at 1.4 W cm$^{-2}$, the surface temperature of Ni-CeO$_{2-x}$/CNTs rapidly rose to 155 °C within 30 s and remained steady (Supplementary Fig. 13). In contrast, no ammonia decomposition was observed at the same temperature in the dark, ruling out any contribution from thermal catalysis alone.

To further distinguish the contributions of photocatalysis and thermo-catalysis, light power dependence experiments and thermal catalytic ammonia decomposition were separately conducted on Ni-CeO$_{2-x}$/CNTs. As shown in Supplementary Fig. 14 and Fig. 2b, when the light intensity increased from 1.4 W cm$^{-2}$ to 2.4 W cm$^{-2}$, the catalyst surface temperature rose from 155 °C to 208 °C, resulting in a nearly six-fold increase in the hydrogen production rate, from 22.3 mmol g$_{cat}^{-1}$ min$^{-1}$ to 128.8 mmol g$_{cat}^{-1}$ min$^{-1}$. However, under the corresponding temperature conditions in the dark thermal catalytic system, the hydrogen production rate remained below 0.1 mmol g$_{cat}^{-1}$ min$^{-1}$. This clearly demonstrates that the reaction is primarily driven by photocatalysis rather than by the light-to-heat effect.

Wavelength dependence experiment of the hydrogen formation rate reveals that, although carbon nanotubes exhibit broad absorption characteristics across the entire UV-visible-near-infrared spectrum (Supplementary Fig. 15), ammonia decomposition over Ni-CeO$_{2-x}$/CNTs occurs significantly only at wavelengths shorter than 600 nm (Fig. 2c). The hydrogen formation rate at different wavelengths closely aligns with the absorption spectrum of CeO$_{2-x}$. This result unambiguously demonstrates that the photoexcited electron−hole pairs generated in CeO$_{2-x}$ under UV-visible light irradiation (≤600 nm) are the key driving force for ammonia decomposition in Ni-CeO$_{2-x}$/CNTs.

The cooperative photothermal strategy enables efficient utilization of the full solar spectrum, leveraging visible light for photoexcitation and infrared light for thermal effects. To validate this concept under realistic conditions, we conducted an outdoor test using a low-cost Fresnel lens to concentrate natural sunlight to an average intensity of 1.08 W cm$^{-2}$ (Fig. 2d). The system achieved a solar-to-hydrogen (STH) efficiency of 3.6%, strongly corroborating the practical potential of our approach for low-cost solar hydrogen production.

To further assess the catalyst's performance under steady-state conditions, we investigated its stability in a continuous-flow reactor. As shown in Fig. 2e, Ni-CeO$_{2-x}$/CNTs exhibited excellent stability during ammonia decomposition under 1.4 W cm$^{-2}$ visible-IR irradiation with a 20% NH$_3$/80% Ar flow (24 mL min$^{-1}$). The hydrogen production rate over Ni-CeO$_{2-x}$/CNTs remained approximately 25 mmol g$_{cat}^{-1}$ min$^{-1}$ over the 55-h continuous-flow conditions. XRD and HRTEM analysis (Supplementary Figs. 16 and 17) confirm that, after long-term testing, the structure of Ni-CeO$_{2-x}$/CNTs and the dispersion of Ni nanoparticles remained intact, with no obvious structural damage or sintering.

We further evaluated the activity of Ni-CeO$_{2-x}$/CNTs at higher light intensity. As shown in Fig. 2f and Supplementary Tables 3 and 4, under full-spectrum light irradiation at 3.5 W cm$^{-2}$ without external heating, the hydrogen production rate of Ni-CeO$_{2-x}$/CNTs reached an impressive 403.8 mmol g$_{cat}^{-1}$ min$^{-1}$ and 298.4 mmol g$_{cat}^{-1}$ min$^{-1}$ under the batch and continuous-flow reaction conditions, respectively. Under 3.5 W cm$^{-2}$ irradiation, a photothermal temperature of 306 °C and a maximum conversion of 92.9% were achieved, which corresponds to a thermodynamic equilibrium temperature of 231 °C. This

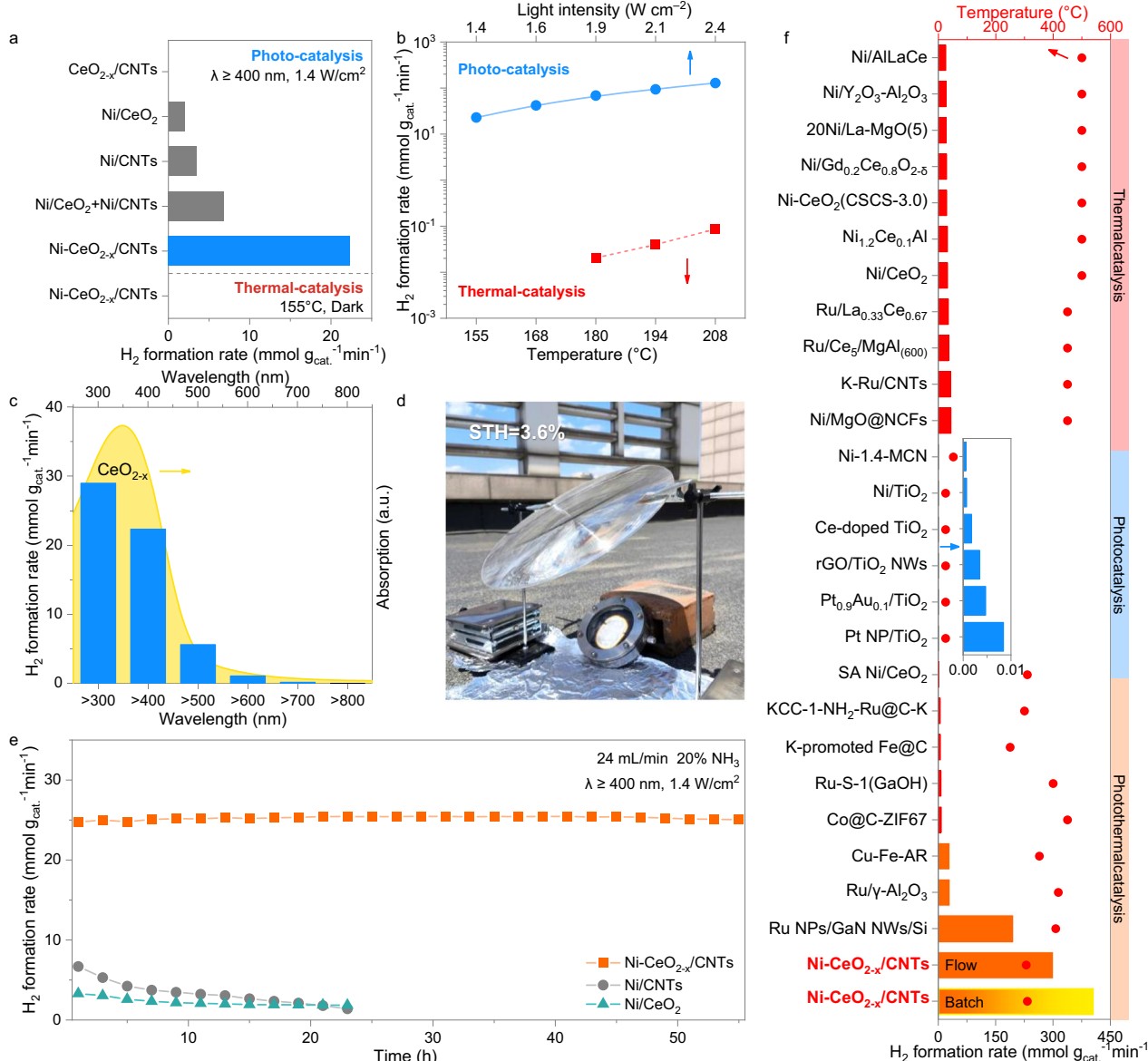

**Fig. 2 | Thermal-assisted photocatalytic ammonia decomposition. a** Reaction performance of various catalysts. **b** Comparison of photocatalysis and thermal catalysis for Ni-CeO$_{2-x}$/CNTs. **c** Wavelength-dependent ammonia decomposition performance of Ni-CeO$_{2-x}$/CNTs and UV-vis DRS of CeO$_{2-x}$. **d** Image of outdoor photocatalytic ammonia decomposition system at Jilin University, Changchun, China. **e** Durability test over Ni-CeO$_{2-x}$/CNTs, Ni/CeO$_2$ and Ni/CNTs. Reaction condition: A mixture of 5 mg catalyst and 100 mg quartz sand, 24 mL min$^{-1}$ 20% NH$_3$/80% Ar, 1.4 W cm$^{-2}$ light irradiation ($\lambda \geq 400$ nm). **f** Summary of H$_2$ formation rate achieved in this work and versus state-of-the-art systems (thermal: red; photo: blue; photo-thermo: orange).

performance, achieved in a continuous-flow setup, is 10 times higher than that of the state-of-the-art Ru-based photothermal catalyst[30] and 9 times higher than that of the best-performing Ni-based thermal catalyst at 450 °C[31].

## The effect of CNTs in boosting photocatalytic performance

Although thermal energy alone cannot drive ammonia decomposition over Ni-CeO$_{2-x}$/CNTs, it significantly accelerates the photocatalytic process. As shown in Fig. 3a, under monochromatic 400 nm light irradiation (1.4 W cm$^{-2}$, cold LED), the catalyst surface temperature reached 74 °C, yielding a hydrogen production rate of 6.7 mmol g$_{cat}^{-1}$ min$^{-1}$. With external heating to 155 °C, the rate sharply increased to 17.5 mmol g$_{cat}^{-1}$ min$^{-1}$, achieving an apparent quantum efficiency (AQE) of 12.7%, the highest value reported to date for photocatalytic ammonia decomposition. Temperature-dependent

photoluminescence (TD-PL) spectroscopy further revealed that elevated temperatures suppress charge recombination, with PL intensity decreasing by over 70% from −150 °C to 150 °C (Fig. 3b)[32]. This enhancement stems from improved charge mobility, supported by electrochemical impedance spectroscopy, which shows a clear reduction in charge transfer resistance with rising temperature (Supplementary Fig. 18)[33].

The introduction of CNTs significantly contributes to the increase in the temperature of the photocatalyst and thus facilitates charge separation[34–36]. As shown in Supplementary Fig. 19, under 920 nm monochromatic light irradiation (1.4 W cm$^{-2}$), the surface temperature of CNTs reached 114 °C, while pure CeO$_2$ only reached 65 °C. This demonstrates the superior light-to-heat conversion capability of CNTs, particularly for the typically wasted near-infrared (NIR) light, which cannot drive photoexcited charge separation in CeO$_{2-x}$.

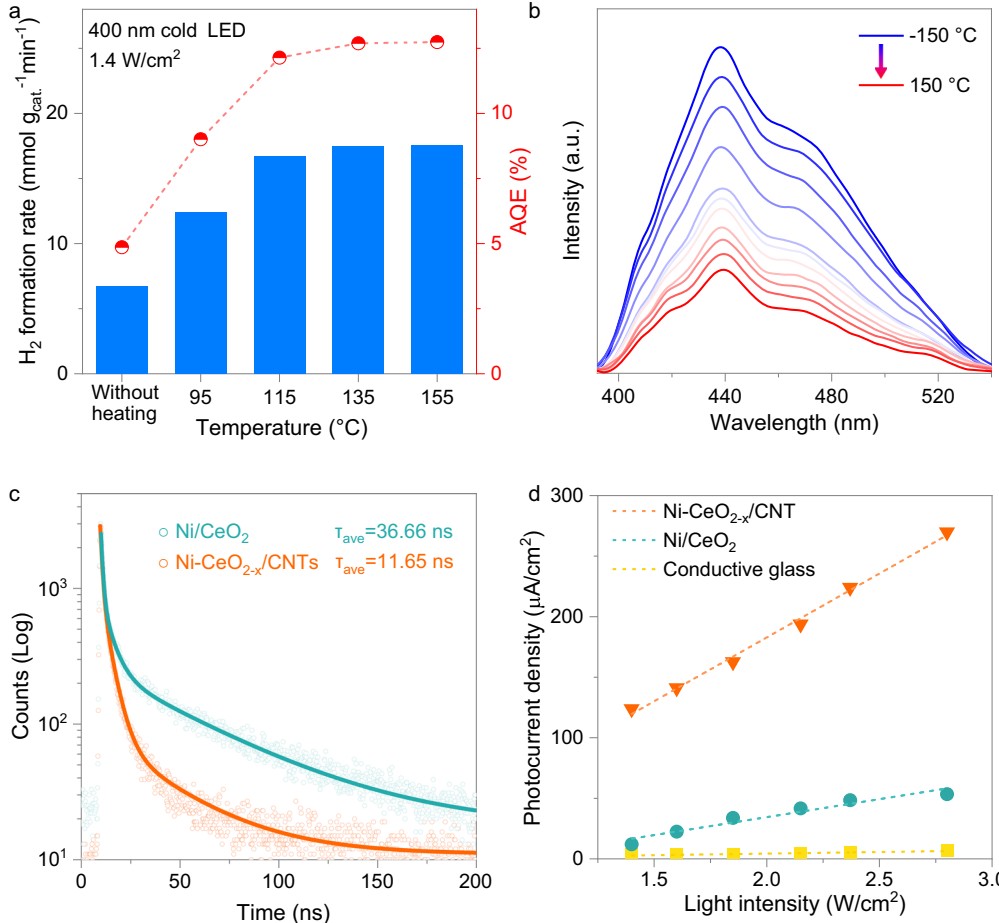

**Fig. 3 | Photoexcited charge dynamics in Ni-CeO$_{2-x}$/CNTs. a** Hydrogen production rate and AQE of Ni-CeO$_{2-x}$/CNTs under monochromatic 400 nm light irradiation without and with external heating. **b** TD-PL spectroscopy of Ni-CeO$_{2-x}$/CNTs at temperatures ranging from −150 °C to 150 °C. **c** TR-PL spectroscopy of Ni-CeO$_{2-x}$/CNTs and Ni/CeO$_2$. **d** Light intensity-dependent photocurrent measurements.

Photocurrent measurements further show that the photocurrent density of Ni-CeO$_{2-x}$/CNTs is significantly higher than that of Ni/CeO$_2$ and Ni/CNTs (Supplementary Fig. 20) even at the same temperature. This enhancement can be attributed to the nanocomposite structure formed by CeO$_{2-x}$ and CNTs, where CeO$_{2-x}$ generates photoexcited electron–hole pairs, and the highly conductive CNTs serve as an "electron transport channel", facilitating rapid interfacial electron migration and effectively suppressing carrier recombination (Supplementary Fig. 21)[37]. The accelerated interfacial charge transfer is further confirmed by time-resolved photoluminescence (TR-PL) spectroscopy. As shown in Fig. 3c and Supplementary Table 5, the introduction of CNTs reduces the average carrier lifetime ($\tau_{ave}$) from 36.66 ns in Ni/CeO$_2$ to 11.65 ns in Ni-CeO$_{2-x}$/CNTs, which is consistent with the enhanced carrier separation observed in carbon-based materials as reported in the literature[38].

Therefore, CNTs contribute to a synergistic effect that significantly enhances the photoexcited carrier separation of Ni-CeO$_{2-x}$/CNTs: their efficient photothermal conversion ability raises the catalyst temperature, reducing carrier transport resistance, while their high electrical conductivity accelerates interfacial electron transfer. As shown in Fig. 3d, light intensity-dependent photocurrent measurements reveal that the photocurrent density of Ni-CeO$_{2-x}$/CNTs is not only significantly higher than that of Ni/CeO$_2$, but the rate of increase in photocurrent with light intensity (slope) is also much greater, fully demonstrating the dual-promoting mechanism of CNTs.

## Reaction mechanism

In situ XPS analysis elucidated the charge transfer pathway during the photocatalytic process. As shown in Fig. 4a, light irradiation induces a 0.14 eV negative shift in the Ni 2$p$ binding energy of Ni-CeO$_{2-x}$/CNTs, indicating that photogenerated electrons from CeO$_{2-x}$ migrate and accumulate on the Ni nanoparticles. Simultaneously, an increase in Ce$^{4+}$ concentration suggests that photogenerated holes remain in CeO$_{2-x}$. Photocurrent measurements reveal a sharp rise in photocurrent density from 42 μA cm$^{-2}$ (under Ar) to 134 μA cm$^{-2}$ upon NH$_3$ introduction (Fig. 4b), confirming that the holes are predominantly consumed by NH$_3$ via oxidative dehydrogenation (NH$_3$ + $h^+$ → NH$_2$ + H$^+$). EPR spectroscopy (Supplementary Fig. 22) further indicates that oxygen vacancies in CeO$_{2-x}$ serve as hole-trapping centers (O$_V$ + $h^+$ → O$_V^+$), playing a crucial role in NH$_3$ activation.

DFT calculations further demonstrate that oxygen vacancies in CeO$_{2-x}$ significantly enhance the adsorption of NH$_x$ (x = 3, 2, 1, 0) at the intimately contacted Ni-CeO$_{2-x}$ interface. As shown in Fig. 4c, the d-band center of interfacial Ni in the oxygen-deficient Ni$_8$/CeO$_2$_O$_V$ model shifts upward to −1.03 eV, compared to −2.00 eV in the vacancy-free Ni$_8$/CeO$_2$ and −1.17 eV in the Ni(111) surface. Consequently, as illustrated in Supplementary Fig. 23, vacancy-induced Ce–O$_V$–Ni interface in Ni$_8$/CeO$_2$_O$_V$ provides enhanced N adsorption sites with an adsorption energy of −1.26 eV, surpassing those of Ni$_8$/CeO$_2$ (−0.49 eV) and Ni(111) (−0.95 eV). This upward shift improves the electronic coupling between the Ni d-states and the LUMO of N*, thereby promoting the coupling of N−N on the catalyst surface[39,40].

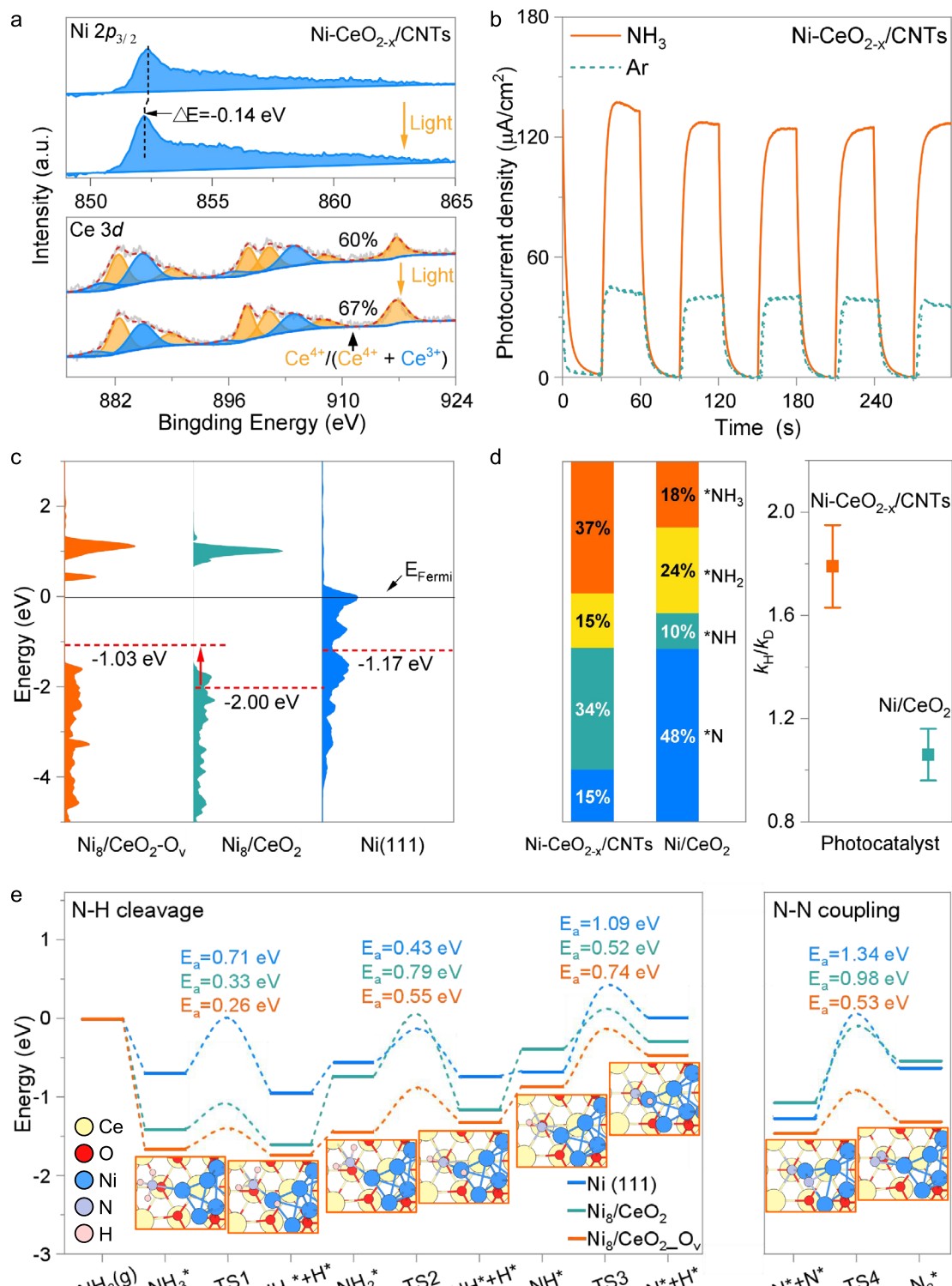

**Fig. 4 | Reaction mechanism. a** In situ XPS spectra of Ni $2p_{3/2}$ and Ce $3d$ on Ni-CeO$_{2-x}$/CNTs under dark conditions and light-irradiated (10 min) conditions. **b** Transient photocurrent responses under light irradiation with/without NH$_3$. **c** The total density of states of Ni$_8$/CeO$_2$_O$_v$, Ni$_8$/CeO$_2$ and Ni(111). The red dashed line indicates the position of the d-band center. **d** The proportion of individual peaks assigned to NH$_x$ (where x = 0, 1, 2, or 3) in the in situ XPS spectra of Ni-CeO$_{2-x}$/CNTs after the ammonia decomposition reaction (right) and Kinetic isotope effects of Ni-CeO$_{2-x}$/CNTs and Ni/CeO$_2$ (left). **e** Energy profiles for NH$_3$ decomposed on Ni(111), Ni$_8$/CeO$_2$ and Ni$_8$/CeO$_2$_O$_v$ surfaces. The optimized structure of each intermediate is depicted, wherein Ce, O, Ni, N, and H atoms are depicted as yellow, red, blue, light blue, and white spheres, respectively.

In traditional nickel-based catalysts, N−H bond cleavage occurs relatively easily, while the N−N coupling barrier is very high (rate-determining step), leading to the accumulation of N* species on the catalyst surface. This accumulation inhibits the catalytic cycle and becomes the main bottleneck limiting decomposition efficiency. The distribution of nitrogen species on the surface of Ni-CeO$_2$ and Ni-CeO$_{2-x}$/CNTs catalysts after reaction was analyzed using XPS N 1s spectra. As shown in Fig. 4d and Supplementary Fig. 24 and

Supplementary Table 6, 48% of the surface nitrogen on Ni-CeO$_2$ remained as atomic N* after reaction, indicating that N atoms tend to accumulate and poison the surface. In contrast, Ni-CeO$_{2-x}$/CNTs showed a markedly reduced N* fraction of only 15%, suggesting that N–N coupling proceeded smoothly.

The change in the rate-determining step of ammonia decomposition over Ni-CeO$_{2-x}$/CNTs is further confirmed through kinetic isotope effect (KIE, $k_H/k_D$) experiments. As shown in Fig. 4d, when ND$_3$ replaced NH$_3$ in the ammonia decomposition reaction, the KIE value measured on Ni/CeO$_2$ was $1.0 \pm 0.1$, confirming that the rate-determining step in traditional nickel-based catalysts is N–N coupling, not N–H bond activation. In contrast, the KIE value for Ni-CeO$_{2-x}$/CNTs reached $1.8 \pm 0.2$, suggesting that N–N coupling is no longer the rate-determining step and has shifted to N–H bond activation[41,42]. Ni-CeO$_{2-x}$/CNTs exhibits kinetic characteristics markedly different from those of conventional Ni-based catalysts. As shown in Supplementary Fig. 25, the corresponding reaction orders were determined to be 0.72 for Ni-CeO$_{2-x}$/CNTs, 0.54 for Ni/CNTs, and 0.44 for Ni/CeO$_2$. The higher reaction order observed for Ni-CeO$_{2-x}$/CNTs, indicates a stronger dependence of the rate on NH$_3$ concentration. This result aligns well with the pronounced KIE values previously observed, together providing robust evidence that N–H bond dissociation constitutes the critical kinetic step in ammonia decomposition. Therefore, the highly efficient photo-driven N–H bond cleavage and rapid N–N coupling enable unprecedented ammonia decomposition efficiency, effectively overcoming the intrinsic limitations of traditional nickel-based catalysts.

Reaction pathway calculations provide a comprehensive overview of the ammonia decomposition process (Fig. 4e and Supplementary Fig. 26). The ammonia decomposition reaction ($2\,NH_3 \rightarrow N_2 + 3H_2$) on the catalyst surface proceeds through several steps: NH$_3$ adsorption, N–H bond cleavage (NH$_3$* $\rightarrow$ NH$_2$* $\rightarrow$ NH* $\rightarrow$ N*), and N–N coupling (2N* $\rightarrow$ N$_2$* $\rightarrow$ N$_2$(g)). As shown in Supplementary Fig. 27, during the NH$_3$ adsorption step, ammonia preferentially adsorbs on the Ni$_8$ cluster surface in the Ni$_8$/CeO$_2$ model. However, in the Ni$_8$/CeO$_2$_O$_V$ model, the Ce–O$_V$–Ni interface exhibits stronger NH$_3$ adsorption, making it the active site for ammonia decomposition. NH$_3$ adsorbed at the Ce–O$_V$–Ni interface undergoes a deprotonation process to form NH$_x$ (x = 2, 1, 0) intermediates. Notably, for the Ni$_8$/CeO$_2$_O$_V$ interface, the *NH $\rightarrow$ *N + *H step shows a relatively high barrier (0.74 eV), though still lower than the highest deprotonation barriers on Ni(111) (1.09 eV) and Ni$_8$/CeO$_2$ (0.79 eV). Simultaneously, N* remains at the Ce–O$_V$–Ni site and recombines with another N* to form N$_2$. For conventional nickel-based catalysts, the relatively weak Ni–N bonds lead to high N–N coupling energy barriers (1.34 eV and 0.98 eV for Ni(111) and Ni$_8$/CeO$_2$, respectively). However, the enhanced Ni–N bond strength at the Ce–O$_V$–Ni interface significantly lowers the N–N coupling energy barrier to 0.53 eV in Ni$_8$/CeO$_2$_O$_V$. Finally, the electron-rich Ni$_8$ can efficiently donate electrons to the N$_2$ π* orbitals, facilitating nitrogen desorption.

## Discussion

The Ni/CeO$_{2-x}$-CNTs catalyst developed in this study marks a significant advancement in the field of photothermal catalysis. It integrates the photocatalytically active CeO$_{2-x}$ nanodomains and electron-rich Ni nanoparticles with intimate interfacial contact on carbon nanotubes. By designing highly efficient interface active sites and combining thermally promoted photocatalytic synergies, this catalyst exhibits exceptional ammonia decomposition efficiency, catalytic stability, and low-barrier reaction kinetics under light irradiation without the need for external heating. The introduction of CNTs significantly enhances photothermal conversion and electron transport, facilitating efficient N–H bond cleavage. The Ce–O$_V$–Ni interface, with an elevated d-band center, strengthens the electronic coupling between the Ni d-states and the LUMO of N, dramatically lowering the

N–N coupling barrier and overcoming the traditional limitations of Ni-based catalysts. This work provides valuable insights into the design of high-performance photothermal catalysts for ammonia decomposition and underscores the potential of thermally assisted photocatalysis, utilizing the full UV-Vis-NIR spectrum of solar energy to create sustainable catalytic solutions powered by clean solar energy.

## Methods

### Preparation of CeO$_{2-x}$/CNTs support

The CeO$_{2-x}$/CNTs support was synthesized using the precipitation method. In a typical procedure, 190 mg of Ce(NO$_3$)$_3$·6H$_2$O and 150 mg of multi-walled carbon nanotubes (MWCNTs, with an outer diameter of 10–20 nm and length of 10-30 μm, surface area >120 m$^2$ g$^{-1}$, provided by Shenzhen Co., Ltd.) were dispersed in 100 mL deionized water. The resulting mixture was stirred for 12 h, followed by the addition of ammonia solution until pH = 9.0, with vigorous stirring for 30 min. The precipitate was washed three times each with deionized water and ethanol, then dried overnight in a vacuum oven at room temperature. Subsequently, the obtained powder was calcined at 873 K for two hours in a tubular furnace under a nitrogen atmosphere (heating rate: 5 K min$^{-1}$; nitrogen flow rate: 60 mL min$^{-1}$). The synthesized sample with a mass ratio of CeO$_2$ to MWCNTs of 0.5 will be referred to simply as CeO$_{2-x}$/CNTs throughout the manuscript. In this study, we prepared a series of samples with varying mass ratios of CeO$_2$ to MWCNTs, including 10 wt%CeO$_{2-x}$/CNTs, 30 wt%CeO$_{2-x}$/CNTs, 100 wt%CeO$_{2-x}$/CNTs, 200 wt%CeO$_{2-x}$/CNTs, 1000 wt%CeO$_{2-x}$/CNTs, and 2000 wt%CeO$_{2-x}$/CNTs, along with the CeO$_{2-x}$/CNTs sample described above. CeO$_{2-x}$ was also synthesized following the same procedure but without incorporating CNTs.

### Preparation of Ni-supported catalysts

The nickel-loaded catalysts were prepared using the chemical reduction method. In a typical procedure, 100 mg of CeO$_{2-x}$/CNTs, commercial CeO$_2$, and CNTs support were individually dispersed in a solution containing 50 mg Ni(NO$_3$)$_2$·6H$_2$O dissolved in 100 mL deionized water. The mixture was stirred overnight in the dark to ensure complete dispersion and adsorption. Thereafter, 50 mg NaBH$_4$ was rapidly added to the solution under vigorous stirring for 30 min. The resulting suspension was filtered, and the collected precipitate was washed three times each with deionized water and ethanol. Finally, the product was dried in a vacuum oven at room temperature. The synthesized samples were named Ni-CeO$_{2-x}$/CNTs, Ni/CeO$_2$, and Ni/CNTs.

### Characterization

Elemental composition was determined through inductively coupled plasma optical emission spectroscopy (ICP-OES) measurements executed on a Thermo Scientific iCAP 7600 system. Powder X-ray diffraction (XRD) analysis was conducted on a Rigaku D/Max 2550 diffractometer employing monochromated Cu Kα radiation (λ = 1.5406 Å), with data acquisition parameters set to 10° min$^{-1}$ scan rate across a 2θ angular range of 20–60°. Microstructural characterization was performed using a Thermo Scientific Talos F200S G2 transmission electron microscope operating at 200 kV, which provided both Transmission electron microscopy (TEM) and high-angle annular dark-field scanning TEM (HAADF-STEM). Scanning electron microscopy (SEM) and energy dispersive X-ray spectroscopy (EDS) analysis were obtained using a Helios NanoLab 600l from FEI Company. Surface area quantification was implemented through nitrogen adsorption isotherms at 77 K (Micromeritics ASAP 2020 M), with specific surface areas calculated via the Brunauer-Emmett-Teller (BET) method and pore size distributions derived from Barrett-Joyner-Halenda (BJH) analysis. In situ X-ray photoelectron spectroscopy (XPS) was analyzed by Thermo Scientific ESCALAB 250Xi with Al Kα excitation (1486.6 eV), where binding energy calibration referenced the adventitious carbon 1s peak at 284.8 eV using Avantage processing

software. Raman spectra were carried out using the Raman Microscope DXR3 with incident radiation at 532 nm (diode lasers of energies 2.33 eV). Ultraviolet photoelectron spectroscopy (UPS) was carried out by a Prevac spectrometer with a VG Scienta R3000 hemispherical electron energy analyzer. UV-vis diffuse reflectance spectroscopy (UV-vis DRS) was tested in the range of 200–2000 nm using PerkinElmer Lambda 950. Electrochemical characterization involved photocurrent measurements using a CHI 660E workstation in a three-electrode configuration (FTO working electrode, Pt counter electrode, Ag/AgCl reference). Photoluminescence (PL), Temperature-dependent PL (TD-PL), and Time-resolution PL (TR-PL) spectroscopies were conducted on an Edinburgh Instruments FLS1000 spectrometer. In situ electron paramagnetic resonance (EPR) spectroscopy (JEOL JES-FA 200) operated at 9.43 GHz with 337 mT center field and 0.998 mW microwave power.

### Performance evaluation

Under batch reaction conditions, photocatalysis was conducted in the customized quartz reactor chamber designed to enable uniform illumination and precise temperature control (Supplementary Fig. 11). 5 mg of catalyst was uniformly dispersed as a thin film onto the quartz reactor base. Then the reactor chamber underwent vacuum degassing (<0.1 mbar) at 473 K for 2 h to rigorously eliminate surface-adsorbed contaminants. After cooling to ambient temperature (298 K), 0.5 mmol of ammonia gas ($NH_3$, 99.99% purity) was introduced into the reactor. The catalyst was irradiated using a 300 W xenon lamp (CEL-HXF300) equipped with a 400 nm cut-off filter. The intensity of the incident light at the catalyst surface was detected using an optical power meter (CEL-NP2000-10). Reaction temperature was regulated using a closed-loop refrigerated/heated circulating bath connected to the reactor jacket. Catalyst surface temperature was monitored by an infrared camera and validated by an embedded thermocouple positioned near the catalyst layer. The products were sampled at defined intervals and quantified using a gas chromatograph (GC-2014, Shimadzu) equipped with a Thermal Conductivity Detector (TCD).

Photocatalytic performance under natural sunlight was evaluated using a setup similar to that employed in batch experiments. Sunlight was concentrated using a Fresnel lens (30 cm in diameter) to an 8 cm diameter spot on the reactor, yielding an average light intensity of 1.08 W cm$^{-2}$. 0.5 mmol of $NH_3$ (99.99% purity) was introduced, and the system was irradiated under concentrated sunlight for 1 min.

Under continuous-flow conditions, 5 mg of catalyst was well mixed with 100 mg of quartz sand and evenly spread on a sintered quartz plate in a custom-designed quartz reactor. The reaction was performed under irradiation from a 300 W Xe lamp (CEL-HXF300) with a 400 nm cut-off filter, using a gas feed of $NH_3$ at a flow rate of 24 mL min$^{-1}$. The gas products were continuously analyzed by gas chromatography (FL9790).

## Data availability

The authors declare that all data supporting the findings of this study are available within the article and Supplementary Information files. All data is available from the corresponding author upon request.

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

## Acknowledgements

The authors are thankful for the financial support by the National Natural Science Foundation (NSFC) of China (Grant No. 22379050 and Grant No. 92577123), National Key Research and Development Program of China (2023YFA1506300), the Science and Technology Development Plan Project of Changchun, China (2024GZZ01), the Natural Science Foundation of Jilin Province (20240302098GX), Doctoral Research Innovation Capability Enhancement Program of Jilin Province (JJKH20250068BS) and the Fundamental Research Funds for the Central Universities.

## Author contributions

R.T. performed the catalyst synthesis, characterization, DFT calculations, and catalytic tests. X.M., Y.K., Q.X., and Q.Z. participated in the characterization. X.W. and Q.J. participated in the catalyst preparation and test. X.M. and L.L. designed the study, analyzed the data, and wrote the paper.

## Competing interests

The authors declare no competing interests.
