## [Peer Review File · Nature Communications]

Breakthrough Photothermal Ammonia Decomposition via Low-Barrier Ni-CeO_{2-x} Interfaces on Carbon Nanotubes

Corresponding Author: Professor Lu Li

Version 0:

Reviewer comments:

Reviewer #1

(Remarks to the Author)

Tan et al. reported a Ni-CeO_{2-x}/CNTs photothermal catalyst that delivers exceptionally high H₂ production rates under light without external heating and claims >91% NH₃ conversion at 155 °C with strong cycling stability. The mechanistic picture attributes the performance to (i) efficient light harvesting/photothermal conversion by CNTs, (ii) photo-assisted N-H activation on CeO_{2-x}, and (iii) a Ce-OV-Ni interfacial site that upshifts the Ni d-band center and lowers the N-N coupling barrier (DFT), supported by UPS/XPS/EPR, photocurrent, and (TR)PL. The topic is timely for low-temperature NH₃ cracking and will be of interest to readers in the fields of catalysis, photothermal chemistry, and hydrogen carriers. My comments are primarily related to the DFT calculations since my research background is associated with computational catalysis. Attached below are the major comments to be addressed before this manuscript can be recommended for publication.

1. The title of this manuscript includes “barrierless interfaces,” yet the DFT results indicate a reduced but finite N-N coupling barrier (i.e., 0.65 eV) at Ni₈/CeO₂_Ov sites (vs ≥1.10 eV on Ni). The authors should avoid overstatement in the title/abstract/results: e.g., “near-barrierless with respect to conventional Ni” or “drastically reduced barrier.”
2. The authors should also include the benchmarking test on some key results, considering a relatively low cut-off energy was selected for all the calculations: “cut-off energy of the plane wave basis was set to 400 eV”. Benchmarking testing on convergence is necessary.
3. How were the holes modelled in these reactions, considering VASP cannot sufficiently model the localized positive/negative charges in the systems; however, the authors stated that the computed energetics are for the reactions such as “NH₃ + h₊ → NH₂ + H⁺”?
4. The authors should compute the transition state energies for the multiple NH_x deprotonation reaction steps. From Fig. 4e, the deprotonation of *NH_x to *N is the step with the largest energy change, which might indicate the most significant kinetic barriers. Therefore, it is not sufficient to conclude based solely on the TS energies of *N-*N coupling to *N₂.

Reviewer #2

(Remarks to the Author)

This manuscript reports a Ni-CeO_{2-x}/CNTs composite photothermal catalyst for ammonia decomposition, achieving a record-breaking hydrogen production rate of 403.8 mmol g_{cat}⁻¹ min⁻¹ under full-spectrum light irradiation—2.1 times higher than that of the state-of-the-art Ru-based photo-thermal catalyst. The catalytic performance is impressive, particularly in overcoming the high-temperature limitations typically associated with conventional Ni-based catalysts. Supported by both experimental data and DFT calculations, the authors convincingly elucidate the synergistic photothermal mechanism in promoting both N-H bond activation and N-N coupling.

The manuscript is clearly written, well-organized, and the results are presented with high scientific rigor. I recommend this manuscript for publication in Nature Communications after the authors address the following comments:

1. The authors have demonstrated the effectiveness of the alkaline precipitation method for constructing highly dispersed, defect-rich CeO_{2-x} nanodomains on CNTs. The question is whether this synthetic strategy possesses general applicability to other commonly supports, such as activated carbon (AC) or MgO. Could the author comment on the application potential for forming small and oxygen-vacancy-rich CeO_{2-x} domains on the support?
2. The excellent catalytic performance under simulated sunlight (Xe lamp) is promising. For broader relevance and potential real-world applicability, the authors are encouraged to conduct validation experiments under natural sunlight, including the recording of key parameters such as light intensity, temperature, and solar-to-hydrogen (STH) efficiency. These data would

strengthen the practical impact of the study.

3. The manuscript emphasizes the critical role of CNTs in enhancing electron transport and facilitating photogenerated charge separation. To further support this, additional electronic band structure analysis—particularly focusing on the nature of the CeO_{2-x}/CNT interface (e.g., Schottky junction or S-scheme heterojunction)—would help elucidate the interfacial charge transfer mechanism.

4. Complementary investigation into the reaction order with respect to NH₃ partial pressure would establish a more robust kinetic foundation, thereby reinforcing the KIE interpretation.

5. The significant reduction in the N–N coupling barrier (from 1.34 eV to 0.65 eV) as revealed by DFT is an important mechanistic insight. It would be beneficial to complement this with experimental determination of the apparent activation energy (E_a) for ammonia decomposition over both Ni/CeO₂ and Ni–CeO_{2-x}/CNTs. This would provide a more complete picture of the intrinsic activity enhancement brought about by the interfacial engineering.

Reviewer #3

(Remarks to the Author)

In this work, Tan and collaborators report a novel photothermal ammonia decomposition catalyst based on Ni–CeO_{2-x} supported on carbon nanotubes. By integrating defect-rich CeO_{2-x} nanodomains and electron-rich Ni nanoparticles on carbon nanotubes, the system achieves highly efficient light-assisted hydrogen production, reaching a remarkable hydrogen generation rate of 403.8 mmol g_{cat}⁻¹·min⁻¹. The enhanced activity arises from synergistic N–H bond activation by photocatalysis and barrierless N–N coupling enabled by Ce–O_v–Ni interfaces. The work could be of interest in the field of photo-thermal catalysis, particularly for NH₃ cracking. However, from my point of view, authors did not provide a fair and accurate comparison of their results from the current state-of-the-art, thus overestimating the real impact of their own work. Furthermore, the experimental methodology should be improved, in particular those related with the stability.

For all these reasons, I cannot recommend the publication of this work in Nature Communications. Detailed comments to support this decision can be found below:

- a) Authors wrote: “Herein we develop the most efficient photo-thermal catalyst to date”. I think this statement should be somehow rephrased as this manuscript lacks relevant references in the field to ensure a meaningful comparison. Authors should make a distinction between batch and continuous-flow reactors, otherwise the comparison is not appropriate.
- b) In line with my previous comment, it is quite surprising that authors did not refer to any of the works of prof. Gascon and colleagues, who were one of the pioneer groups in developing photo-thermal systems for the decomposition of NH₃ under continuous flow configuration (check doi.org/10.1002/cssc.202500068, doi.org/10.1002/cssc.202401896, doi.org/10.1002/sml.202411468)
- c) Authors should clarify if they use visible or visible-IR radiation. As per $\lambda \geq 400$ nm is not clear.
- d) Authors perform a series of 100 reaction cycles to assess the stability of the system. At first this number of cycles could seem significant, but taking into account that the reaction time is one minute, this stability test represents a total reaction time below 2 hours. A longer stability test (at least 50 hours) is needed in order to properly evaluate the integrity of this system with others already available in the literature, especially taking into account that the performance shows a subtle but steady deactivation upon reuses. Authors should also include the complete characterization of the long-term spent sample.
- e) In line with my previous comment, why using a total reaction time of 1 minute? This is an extremely short time which makes very difficult to assess stability or the kinetics of the system. Authors have to increase the initial amount of NH₃, as now is limited to only 12 mL. This will also demonstrate the applicability of the system.
- f) Most of the works on photo-thermal ammonia decomposition have been performed under continuous-flow configuration, which is closer to real application. Authors should perform their activity tests under these conditions to contextualize their results.
- g) In line 173, authors claim that the conversion surpasses the thermodynamic equilibrium. As we cannot trick thermodynamics, this means that the actual temperature of the active sites is higher than that registered by the thermocouple (155 degrees Celsius). Authors claim that the temperature was also monitored by IR but there aren't any IR thermal images neither in the main manuscript nor in the SI. I suggest authors to back-calculate the actual temperature of the active sites in their system, for instance using their actual conversion at the equilibrium. See the work from prof. Zhang (doi.org/10.1002/anie.202304452)

Version 1:

Reviewer comments:

Reviewer #1

(Remarks to the Author)

A minor comment: while most figures are high-resolution vector graphics, the atomic configuration panels, particularly Fig. 4E, make it difficult to distinguish among intermediates. I suggest that the authors enhance the resolution of these images. Aside from this, the authors have adequately addressed all of my comments, and I am pleased to recommend the manuscript for publication in Nature Communications.

Reviewer #2

(Remarks to the Author)

This manuscript can be accepted by NC.

Reviewer #3

(Remarks to the Author)

I appreciate the time and effort of the authors to address my comments. Overall, I think the manuscript improved its clarity after this revision.

Regarding my question about reaction time, now it is clear that authors used 1 min reaction time to measure initial reaction rates. And the actual conversion of those experiments is 91 % after 20 minutes of reaction (as per new figure R12). Still it is quite surprising that the catalyst is able to achieve a 92 % conversion under continuous-flow (24 mL/min flow rate of pure NH₃) only at 300 degrees Celsius. Specially taking into account that under batch conditions a total reaction time of 20 min was necessary to convert 91 % of 12 mL of NH₃.

Authors declined to perform back-calculation of reaction temperature using equilibrium conversions, but probably these data could be useful to estimate the real temperature, which most likely will be higher than 300 degrees Celsius.

Responses (R) to Referees' Comments (C)

We thank all the Reviewers for their comments on the manuscript.

Referee 1

1. The title of this manuscript includes "barrierless interfaces," yet the DFT results indicate a reduced but finite N–N coupling barrier (i.e., 0.65 eV) at Ni₈/CeO₂_Ov sites (vs ≥ 1.10 eV on Ni). The authors should avoid overstatement in the title/abstract/results: e.g., "near-barrierless with respect to conventional Ni" or "drastically reduced barrier."

R: We sincerely thank the reviewer for this valuable comment. We agree that the use of "barrierless" is an overstatement, as the DFT results show that the N–N coupling barrier, while drastically reduced, still remains non-negligible. We have revised the title to: **"Breakthrough Photothermal Ammonia Decomposition via Low-Barrier Ni-CeO_{2-x} Interfaces on Carbon Nanotubes."** Throughout the manuscript, "barrierless" has been replaced with more accurate terms such as "significantly reduced barrier" or "rapid reaction kinetics."

2. The authors should also include the benchmarking test on some key results, considering a relatively low cut-off energy was selected for all the calculations: "cut-off energy of the plane wave basis was set to 400 eV". Benchmarking testing on convergence is necessary.

R: We sincerely thank the reviewer for this important methodological comment regarding the plane-wave cutoff energy. In our initial calculations, we selected a cutoff energy of 400 eV, balancing efficiency and accuracy, following several prior studies on CeO₂-based systems for ammonia decomposition (10.1002/aenm.202202459; 10.1002/anie.202501898; 10.1016/j.cej.2023.145371). However, we fully acknowledge that not performing convergence tests for the specific systems in this study was inadequate, and we apologize for this oversight.

Accordingly, we have now conducted a systematic convergence test. For both the Ni₈/CeO₂ and Ni₈/CeO₂_Ov models, the results (Table R1–2) indicate that a cutoff energy of 550 eV is required to converge the total energy within 1 meV per atom. This value is consistent with the recommended $1.3 \times \text{ENMAX}$ (400 eV) guideline. Therefore, we have recalculated all results related to Ni₈/CeO₂ and Ni₈/CeO₂_Ov using a cutoff energy of 550 eV, and updated Fig. 1G, Fig. 4C and E, as well as Supplementary Figs. 9, 10, 23, 26, and 27 in the revised manuscript.

Table R1. Convergence test of Ni₈/CeO₂_Ov model

Cut-off energy (eV)	Energy (eV)	Δ Energy (meV/atom)
300	-1298.71258	
350	-1291.36366	-48.67
400	-1287.97054	-22.47
450	-1286.39880	-10.41
500	-1285.82940	-3.77
550	-1285.69113	-0.92

Table R2. Convergence test of Ni₈/CeO₂ model

Cut-off energy (eV)	Energy (eV)	ΔEnergy (meV/atom)
300	-1304.49842	
350	-1299.56142	-32.48
400	-1296.74989	-18.50
450	-1295.32655	-9.36
500	-1294.78561	-3.56
550	-1294.66622	-0.79

For the Ni(111) surface model, the pseudopotential ENMAX value is 269.532 eV. Combined with the convergence test results (Table R3), this indicates that a cutoff energy of 400 eV is sufficient to achieve energy convergence well below 1 meV per atom. Therefore, we have retained the original parameters for this model, and the corresponding results remain unchanged.

Table R3. Convergence test of Ni(111) model

Cut-off energy (eV)	Energy (eV)	ΔEnergy (meV/atom)
350	-249.89151	
400	-249.84658	-0.94
450	-249.83974	-0.14

It is worth emphasizing that all key conclusions and energy trends remain fully consistent after recalculation. The observed reduction in the N–N coupling barrier and the enhancement of interfacial catalytic activity are unchanged. We have updated the cutoff energy settings in the Computational Methods section of the manuscript and sincerely apologize once again for the earlier oversight in parameter selection. We are deeply grateful to the reviewer, whose comments have significantly improved the rigor and accuracy of our computational analysis.

3. How were the holes modelled in these reactions, considering VASP cannot sufficiently model the localized positive/negative charges in the systems; however, the authors stated that the computed energetics are for the reactions such as “ $\text{NH}_3 + \text{h}^+ \rightarrow \text{NH}_2 + \text{H}^+$ ”?

R: We sincerely thank the reviewer for pointing out this issue in the computational description. The original manuscript incorporated reactions such as “ $\text{NH}_3 + \text{h}^+ \rightarrow \text{NH}_2 + \text{H}^+$ ” with the intention of illustrating the respective roles of photogenerated holes and electrons in the photocatalytic ammonia decomposition reaction. However, we fully acknowledge that placing such descriptions in the computational methodology section may lead to misunderstanding, as VASP does not explicitly model localized positive/negative charges.

To avoid ambiguity, we have removed the descriptions related to photogenerated charge-involved steps (e.g., $\text{NH}_3 + \text{h}^+ \rightarrow \text{NH}_2 + \text{H}^+$ and $\text{H}^+ + \text{e}^- \rightarrow \text{H}$) from the theoretical computational section in the revised manuscript. We deeply appreciate this important clarification, which has significantly improved the accuracy of our computational narrative.

4. The authors should compute the transition state energies for the multiple NH_x deprotonation reaction steps. From Fig. 4e, the deprotonation of *NH_x to *N is the step with the largest energy change, which might indicate the most significant kinetic barriers. Therefore, it is not sufficient to conclude based solely on the TS energies of *N-*N coupling to *N₂.

R: We thank the reviewer for this valuable suggestion. In our initial work, we followed prior studies (10.1002/aenm.202202459; 10.1016/j.cej.2023.145371), where N–N coupling on Ni-based catalysts is generally regarded as the rate-determining step, and transition-state calculations for N–H activation are often omitted. To reduce computational cost, we therefore did not calculate all deprotonation transition states. However, we fully agree with the reviewer’s point that in Fig. 4e, the deprotonation process from *NH_x to *N exhibits a considerable energy change, and its kinetic contribution cannot be neglected.

Following your suggestion, we systematically recalculated the transition states for all deprotonation steps from NH₃ to N (*NH₃ → *NH₂ + *H, *NH₂ → NH + H, *NH → N + H) on the Ni(111), Ni₈/CeO₂, and Ni₈/CeO₂_Ov models (Figure R1). Notably, for the Ni₈/CeO₂_Ov interface, the *NH → *N + *H step shows a relatively high barrier (0.74 eV), though still lower than the highest deprotonation barriers on Ni(111) (1.09 eV) and Ni₈/CeO₂ (0.79 eV). Moreover, as shown in the revised Fig. 4E, the deprotonation barriers on Ni(111) and Ni₈/CeO₂ surfaces remain lower than their respective N–N coupling barriers, while at the Ni₈/CeO₂_Ov interface the deprotonation barrier, although significantly reduced, still exceeds the corresponding N–N coupling barrier.

This new finding further confirms that for highly active Ni-based catalysts, the rate-determining step shifts from N–N coupling to N–H bond cleavage, in agreement with our in situ XPS and KIE experimental results. We have revised the relevant discussion in the manuscript to incorporate these computational results, ensuring greater comprehensiveness and accuracy. We are once again grateful to the reviewer for helping us strengthen the scientific rigor of this study.

Figure R1. Energy profiles for NH₃ decomposed on Ni(111), Ni₈/CeO₂ and Ni₈/CeO₂_Ov surfaces (Revised Fig. 4E).

Referee 2

1. The authors have demonstrated the effectiveness of the alkaline precipitation method for constructing highly dispersed, defect-rich CeO_{2-x} nanodomains on CNTs. The question is whether this synthetic strategy possesses general applicability to other commonly supports, such as activated carbon (AC) or MgO. Could the author comment on the application potential for forming small and oxygen-vacancy-rich CeO_{2-x} domains on the support?

R: We sincerely thank the reviewer for this important question. Following your suggestion, we applied the same alkaline precipitation method on activated carbon (AC) and MgO supports. TEM images (Figure R2) show severe particle agglomeration on both AC and MgO, unlike the highly dispersed, nanoscale CeO_{2-x} domains obtained on CNTs. This indicates that the method strongly depends on the surface chemistry of the support. CNTs, with their sp²-hybridized surface, large surface area, and tunable functional groups, provide effective nucleation sites and suppress overgrowth, enabling the formation of small, oxygen-vacancy-rich CeO_{2-x} domains. In contrast, the disordered porous surface of AC and the highly polar surface of MgO hinder uniform anchoring and dispersion, leading to aggregation.

Figure R2. TEM images of CeO₂/AC (a) and CeO₂/MgO (b)

We fully agree with the reviewer that constructing highly dispersed, defect-rich CeO_{2-x} domains on diverse supports is of great importance. In fact, incorporating rare-earth oxide nanostructures onto conventional supports has been shown to effectively suppress metal sintering, tune electronic structures, and enhance catalytic stability and activity (e.g., Zhou et al., Nat. Mater. 2025; Gao et al., Nature 2024; Li et al., Chem Catal. 2024). Our results suggest that while the alkaline precipitation method works most effectively on CNTs, broadening its applicability through tailored support–oxide interface engineering remains a promising pathway for designing high-performance catalysts.

2. The excellent catalytic performance under simulated sunlight (Xe lamp) is promising. For broader relevance and potential real-world applicability, the authors are encouraged to conduct validation experiments under natural sunlight, including the recording of key parameters such as light intensity, temperature, and solar-to-hydrogen (STH) efficiency. These data would strengthen the practical impact of the study.

R: We sincerely thank the reviewer for this insightful suggestion. Conducting validation experiments under natural sunlight is indeed essential to assess the real-world application potential of the catalyst. To this end, we performed outdoor tests in Changchun under clear-sky conditions to explore the catalyst's practical performance. In the experiment, a Fresnel lens (30 cm in diameter) was used to concentrate sunlight into an 8 cm diameter spot, yielding an average light intensity of 1.08 W/cm². The reactor was a custom-designed flat circular chamber, with its base uniformly coated with a thin catalyst film (Figure R3). After vacuum drying, 0.5 mmol of high-purity NH₃ was introduced into the system, which was then irradiated by natural sunlight for 1 min. The hydrogen produced was quantified using the same online gas chromatography method described in the main text, yielding a total H₂ production of 494.76 μmol.

Figure R3. Image of outdoor photocatalytic ammonia decomposition system at Jilin University, Changchun, China (Revised Fig. 2D).

We further evaluated its solar-to-hydrogen (STH) conversion performance according to the standard formula:

$$\begin{aligned}
 STH &= \frac{\text{Chemical energy output}}{\text{Solar energy input}} \\
 &= \frac{\text{Hydrogen production rate} \times \Delta G_{H_2O \rightarrow H_2 + \frac{1}{2}O_2}}{\text{Light intensity} \times \text{Irradiated area}} \\
 &= \frac{r_{H_2} (\text{mol s}^{-1}) \times \Delta G_{H_2O \rightarrow H_2 + \frac{1}{2}O_2}}{P (\text{W cm}^{-2}) \times S (\text{cm}^2)}
 \end{aligned}$$

where r_{H_2} is the hydrogen evolution rate, ΔG is the Gibbs free energy change for the reaction $H_2O \rightarrow H_2 + 1/2O_2$ (237 kJ mol⁻¹), P is the incident light intensity, and S is the irradiated area. The specific calculation process is as follows:

$$STH = \frac{\frac{494.76 \times 10^{-6}}{60} \text{ mol s}^{-1} \times 237000 \text{ J mol}^{-1}}{1.08 \text{ W cm}^{-2} \times 50 \text{ cm}^2} = 3.6 \%$$

Under natural sunlight, the Ni-CeO_{2-x}/CNTs catalyst achieved a solar-to-hydrogen (STH) efficiency of 3.6%, which surpasses that of most previously reported photocatalytic systems (10.1021/acscenergylett.1c02591). This result demonstrates the catalyst's exceptional capability for efficient hydrogen generation under real sunlight conditions. These findings clearly indicate that the catalyst can drive ammonia decomposition without any external energy input, thereby highlighting its strong potential for practical solar-driven applications.

This result and related methods has been incorporated into the revised manuscript and the corresponding figure (Revised Fig. 2D) has been added.

3. The manuscript emphasizes the critical role of CNTs in enhancing electron transport and facilitating photogenerated charge separation. To further support this, additional electronic band structure analysis—particularly focusing on the nature of the $\text{CeO}_{2-x}/\text{CNT}$ interface (e.g., Schottky junction or S-scheme heterojunction)—would help elucidate the interfacial charge transfer mechanism.

R: We sincerely thank the reviewer for this insightful and critical suggestion. A detailed analysis of the electronic band structure, particularly at the $\text{CeO}_{2-x}/\text{CNT}$ interface, is indeed crucial for elucidating the interfacial charge transfer mechanism. We fully agree that incorporating this analysis substantially strengthens the central argument of our manuscript regarding the role of CNTs in facilitating electron transport and charge separation. We employed a combination of ultraviolet photoelectron spectroscopy (UPS, to determine the work function) measurements to investigate the band structure, XPS valence band (XPS-VB, to determine the position of the Fermi energy level), and UV-vis DRS (to determine the band gap) of $\text{CeO}_{2-x}/\text{CNTs}$, CeO_{2-x} , and CNTs. The detailed results are presented in the figure below (Figure R4-5).

Figure R4. (a) Ultraviolet photoelectron spectroscopy (UPS). (b) XPS valence band (XPS-VB) spectra. (c) UV-vis diffuse reflectance spectra (UV-vis DRS).

Figure R5. The band structure of CeO_{2-x} , CNTs and $\text{CeO}_{2-x}/\text{CNTs}$.

As shown in Figure R5, the work function of CNTs (4.91 eV) is higher than that of CeO_{2-x} (4.67 eV). Upon forming a close interface, electrons spontaneously transfer from CNTs to CeO_{2-x} until their Fermi levels equilibrate, with the composite exhibiting a work function of 4.71 eV. This redistribution of charge bends the bands of CeO_{2-x} upward at the interface, generating an internal electric field directed from CNTs to CeO_{2-x}. Such a built-in field effectively drives photogenerated electrons from CeO_{2-x} toward CNTs while pushing holes in the opposite direction. The Schottky barrier formed between the metallic CNTs and semiconducting CeO_{2-x}, together with the resulting space-charge separation gradient, enables efficient carrier separation and markedly suppresses recombination.

In summary, our analysis confirms the formation of a Schottky junction at the CeO_{2-x}/CNT interface. This mechanism clearly explains the role of CNTs as efficient electron acceptors: the built-in field of the Schottky junction greatly promotes electron separation and extraction. This conclusion is strongly supported by our experimental observations, including shortened carrier lifetimes (TR-PL) and significantly enhanced photocurrent responses, thereby providing compelling evidence for the pivotal role of CNTs in enhancing photocatalytic performance.

4. Complementary investigation into the reaction order with respect to NH₃ partial pressure would establish a more robust kinetic foundation, thereby reinforcing the KIE interpretation.

R: We sincerely thank the reviewer for this valuable suggestion. Investigating the reaction order with respect to NH₃ partial pressure indeed provides a more rigorous kinetic basis for interpreting the kinetic isotope effect (KIE). Following this recommendation, we systematically measured the initial reaction rates of Ni-CeO_{2-x}/CNTs, Ni/CNTs, and Ni/CeO₂ catalysts under varying NH₃ partial pressures.

Figure R6. Dependences of NH₃ decomposition rate on the partial pressures of NH₃ (Revised Supplementary Fig. 25)

As shown in Figure R6, the corresponding reaction orders were determined to be 0.72 for Ni-CeO_{2-x}/CNTs, 0.54 for Ni/CNTs, and 0.44 for Ni/CeO₂. The higher reaction order observed for Ni-CeO_{2-x}/CNTs, approaching unity, indicates a stronger dependence of the rate on NH₃ concentration. This finding suggests that N–H bond cleavage is more likely to be involved in the rate-determining step for this catalyst. Importantly, this conclusion aligns well with the pronounced KIE values previously observed, together providing robust evidence that N–H bond dissociation constitutes the critical kinetic step in ammonia decomposition. This analysis has been incorporated into the revised manuscript, and the corresponding figure has been added (RevisedSupplementary Fig. 25), thereby strengthening the kinetic foundation of our study.

5. The significant reduction in the N–N coupling barrier (from 1.34 eV to 0.65 eV) as revealed by DFT is an important mechanistic insight. It would be beneficial to complement this with experimental determination of the apparent activation energy (E_a) for ammonia decomposition over both Ni/CeO₂ and Ni–CeO_{2-x}/CNTs. This would provide a more complete picture of the intrinsic activity enhancement brought about by the interfacial engineering.

R: We sincerely thank the reviewer for this valuable suggestion. Measuring and comparing the apparent activation energies (E_a) of Ni/CeO₂ and Ni–CeO_{2-x}/CNTs catalysts provides direct experimental evidence for the reduced energy barrier in ammonia decomposition. To address this, we conducted complementary kinetic experiments. Under different light intensities, we measured the corresponding photothermal temperatures and ammonia decomposition rates of the catalysts. By fitting the data with the Arrhenius equation, we successfully determined the apparent activation energies (Figure R7).

The results show that the apparent activation energy of Ni/CeO₂ is 70.3 kJ mol⁻¹, whereas that of Ni–CeO_{2-x}/CNTs is markedly reduced to 47.7 kJ mol⁻¹. From an experimental kinetics perspective, this strongly confirms that the interfacial structure formed by oxygen-vacancy-rich CeO_{2-x} and CNTs effectively optimizes the reaction pathway and significantly lowers the overall energy barrier of ammonia decomposition.

Figure R7. Arrhenius plots of Ni/CeO₂ and Ni–CeO_{2-x}/CNTs catalyst

Referee 3

a) Authors wrote: “Herein we develop the most efficient photo-thermal catalyst to date”. I think this statement should be somehow rephrased as this manuscript lacks relevant references in the field to ensure a meaningful comparison. Authors should make a distinction between batch and continuous-flow reactors, otherwise the comparison is not appropriate.

b) In line with my previous comment, it is quite surprising that authors did not refer to any of the works of prof. Gascon and colleagues, who were one of the pioneer groups in developing photo-thermal systems for the decomposition of NH_3 under continuous flow configuration (check doi.org/10.1002/cssc.202500068, doi.org/10.1002/cssc.202401896, doi.org/10.1002/sml.202411468)

R: We sincerely thank the reviewer for this critical suggestion, which has greatly improved the rigor of our work. We fully agree that our original statement of having developed “the most efficient photo-thermal catalyst to date” was overstated, particularly given the lack of sufficient comparison with different reactor systems, especially continuous-flow configurations. We also sincerely apologize for not citing the pioneering contributions of Prof. Gascon’s group in the field of continuous-flow photothermal ammonia decomposition. Their work has been instrumental in advancing the understanding of photothermal catalysis under flow conditions. We have now cited these studies in the revised introduction (Ref. 18–20).

To address the reviewer’s comment and enable a more rigorous comparison, we performed additional validation experiments under continuous-flow conditions (Figure R8). Specifically, 5 mg of catalyst was mixed with 100 mg of quartz sand and evenly spread on a sintered quartz plate in a custom-designed quartz reactor. Under irradiation with a 300 W Xe lamp (CEL-HXF300, 3.5 W cm^{-2}), high-purity NH_3 (99.99%) was fed at 24 mL min^{-1} (WHSV = $288,000 \text{ mL g}^{-1} \text{ h}^{-1}$). The gas products were continuously analyzed by gas chromatography (FL9790). Under these conditions, the Ni-CeO_{2-x}/CNTs catalyst achieved 92.9% NH_3 conversion with a hydrogen production rate of $298.4 \text{ mmol g}^{-1} \text{ min}^{-1}$.

Figure R8. (a) Image of photo-thermal catalytic device for NH_3 decomposition. (b) Schematic images. (c) Image of the customized quartz reactor.

We have added these results to the revised Supplementary Table 4, together with data from Prof. Gascon’s studies, and clearly indicated the reactor types for each system to ensure a fair comparison. Remarkably, even under flow conditions, Ni-CeO_{2-x}/CNTs maintains a high hydrogen production rate, which is approximately ten times higher than that of the previously reported benchmark photothermal catalyst (Ru/ γ -Al₂O₃). This reinforces the robustness and outstanding performance of our catalyst in both batch and flow systems.

Revised Supplementary Table 4. Summary of various reported catalysts in photo- and photo-thermal catalysis for ammonia decomposition.

Entry	Catalysts	Ammonia source	Light source	Reactor type	T (°C)	r_{H_2} (mmol _{H₂} ·g _{cat} ⁻¹ ·min ⁻¹)	Ref.
1	Ce-doped TiO ₂	Ammonia solution	UV 8 W Hg pen-ray lamp	Batch reactor	r.t.	1.8×10 ⁻³	26
2	rGO/TiO ₂ NWs	Ammonia solution	UV 8 W Hg pen-ray lamp	Batch reactor	r.t.	3.5×10 ⁻³	27
3	Pt _{0.9} Au _{0.1} /TiO ₂	Ammonia solution	2000 W Xe lamp	Batch reactor	r.t.	4.7×10 ⁻³	28
4	Pt NP/TiO ₂	NH ₃ with water vapour	300 W Xe lamp	Batch reactor	r.t.	9.3×10 ⁻³	29
5	Ni-1.4-MCN	5% NH ₃ gas	300 W Xe lamp	Continuous-flow reactor	52	5.9×10 ⁻⁴	30
6	SA Ni/CeO ₂	33% NH ₃ gas	1 sun	Continuous-flow reactor	310	1.6	31
7	KCC-1-NH ₂ -Ru@C-K	NH ₃ gas	Xe lamp	Continuous-flow reactor	300	5.1	32
8	K-promoted Fe@C	NH ₃ gas	300 W Xe lamp	Continuous-flow reactor	250	5.6	33
9	Ru-S-1 (GaOH)	NH ₃ gas (99.999%)	400 W Xe lamp	Continuous-flow reactor	400	7.2	34
10	Co@C-ZIF67	NH ₃ gas	300 W Xe lamp	Continuous-flow reactor	450	8.0	35
11	Cu-Fe-AR	NH ₃ gas (99.99%)	White-light laser	Continuous-flow reactor	352	28.0	36
12	Ru/γ-Al ₂ O ₃	NH ₃ gas (99.999%)	300 W Xe lamp	Continuous-flow reactor	418	28.4	37
13	Ru NPs/GaN NWs/Si	Ammonia solution	300 W Xe lamp	Batch reactor	409	184.3	38
14 ^a	Ni-CeO _{2-x} /CNTs	NH ₃ gas (99.99%)	300 W Xe lamp	Batch reactor	155	22.3	This work
15 ^b	Ni-CeO _{2-x} /CNTs	NH ₃ gas (99.99%)	300 W Xe lamp	Batch reactor	310	403.8	This work
16 ^c	Ni-CeO _{2-x} /CNTs	20% NH ₃ gas	300 W Xe lamp	Continuous-flow reactor	152	25.4	This work
17 ^d	Ni-CeO _{2-x} /CNTs	NH ₃ gas (99.99%)	300 W Xe lamp	Continuous-flow reactor	306	298.4	This work

^a **Batch reaction conditions:** Under standard testing conditions, catalytic ammonia conversion (0.5 mmol) was conducted using 5 mg catalyst under light irradiation (300 W Xe lamp, equipped with a 400 nm cut-off filter) with light density maintained at 1.4 W/cm², duration = 1 min.

^b **Batch reaction conditions:** Catalytic ammonia conversion (0.5 mmol) was conducted using 1 mg catalyst under light irradiation (300 W Xe lamp) with light density maintained at 3.5 W/cm², duration = 1 min.

^c **Continuous-flow reaction conditions:** A mixture of 5 mg catalyst and 100 mg quartz sand was tested with a 20% NH₃/80% Ar flow rate of 24 mL/min, under light irradiation (300 W Xe lamp, equipped with a 400 nm cut-off filter) with light density maintained at 1.4 W/cm².

^d **Continuous-flow reaction conditions:** A mixture of 5 mg catalyst and 100 mg quartz sand was tested with a pure NH₃ flow rate of 24 mL/min, under light irradiation (300 W Xe lamp) with light density maintained at 3.5 W/cm².

In summary, we have made three major revisions to the manuscript in response to the reviewer's comments. First, we have incorporated and highlighted the key contributions of Prof. Gascon's group in this field (Revised References 18–20, Revised Supplementary References 32–33, 35, and Revised Fig. 2F). Second, we have added experimental data on the performance of our catalyst under continuous-flow conditions and summarized these results in Revised Supplementary Table 4, where the reactor configurations of each study are clearly indicated. Finally, we have revised the absolute claim "the most efficient" to "one of the most efficient," providing a more accurate representation of our findings.

We sincerely thank the reviewer once again for these highly constructive comments, which have significantly improved the rigor and comprehensiveness of our work.

c) Authors should clarify if they use visible or visible-IR radiation. As per $\lambda \geq 400$ nm is not clear.

R: We sincerely thank the reviewer for raising this important point regarding the precise description of our light source. In our standard tests, we utilized a 300 W xenon lamp equipped with a 400 nm cut-off filter, providing irradiation spanning the visible to near-infrared spectrum (visible-IR irradiation). To address this concern unequivocally and prevent any potential misunderstanding, we have revised the manuscript accordingly.

Specifically, in Section 2.2 of the revised manuscript, we now clearly state: "Photothermal ammonia decomposition was conducted in a fixed-bed reactor under visible-IR irradiation (300 W xenon lamp equipped with a 400 nm cut-off filter, 1.4 W/cm²) without external heating." Furthermore, all instances of the vague term "visible radiation" have been replaced with the accurate "visible-IR irradiation" throughout the manuscript. The Methods section has also been updated to explicitly state that, unless otherwise specified, all photocatalytic reactions were carried out under illumination from the 300 W xenon lamp with the 400 nm cut-off filter.

We are grateful to the reviewer for this valuable comment, which has significantly enhanced the precision and scientific rigor of our work.

d) Authors perform a series of 100 reaction cycles to assess the stability of the system. At first this number of cycles could seem significant, but taking into account that the reaction time is one minute, this stability test represents a total reaction time below 2 hours. A longer stability test (at least 50 hours) is needed in order to properly evaluate the integrity of this system with others already available in the literature, especially taking into account that the performance shows a subtle but steady deactivation upon reuses. Authors should also include the complete characterization of the long-term spent sample.

R: We sincerely thank the reviewer for this crucial suggestion. In response, and to comprehensively evaluate the catalyst's stability, we conducted a continuous-flow stability test for over 50 hours using the flow reactor system (Figure R8). The experimental conditions were as follows: 5 mg of catalyst was mixed with 100 mg of quartz sand and placed in the reactor. The system was irradiated with a 300 W xenon lamp (equipped with a 400 nm cutoff filter, light intensity 1.4 W/cm²), and a 20% NH₃/Ar mixture was passed through at a flow rate of 24 mL/min (WHSV = 57,600 mL g⁻¹ h⁻¹).

The results (Figure R9) show that the Ni-CeO_{2-x}/CNTs catalyst maintained a stable hydrogen production rate of approximately 25 mmol g_{cat.}⁻¹ min⁻¹ over the 55-hour continuous reaction, demonstrating excellent catalytic stability. Furthermore, we conducted comprehensive microscopic structural characterization of the spent catalyst (long-term used sample). X-ray diffraction (XRD) and aberration-corrected HAADF-STEM-EDS analysis (Figures R10-R11) confirmed that, after long-term testing, the dispersion of Ni nanoparticles on CNTs remained perfectly intact, with no observable sintering or particle growth. In contrast, the Ni nanoparticles on Ni/CNTs aggregated, which contributed to the observed decrease in hydrogen production rate. These characterization results provide direct structural evidence supporting the outstanding stability of our catalyst.

Figure R9. Durability test over Ni-CeO_{2-x}/CNTs, Ni/CeO₂ and Ni/CNTs (Revised Fig. 2E)

Figure R10. Powder XRD patterns for Ni-CeO_{2-x}/CNTs before and after the 55-h stability test (post-test sample mixed with quartz sand) (Revised Supplementary Fig. 16)

Figure R11. HRTEM images and size distribution of Ni nanoparticles taken from Ni-CeO_{2-x}/CNTs before (a-c) and after (d-f, Revised Supplementary Fig. 17) the 55-hour stability test.

This valuable suggestion from the reviewer has prompted us to elevate the stability tests to a more convincing and comparable standard. These new data greatly strengthen our conclusions regarding the catalyst's durability, and we have fully integrated them into the Results and Discussion sections of the revised manuscript (Revised Fig. 2E and Revised Supplementary Fig. 16-17). Once again, we sincerely thank you for helping us significantly improve the quality and rigor of this study.

e) In line with my previous comment, why using a total reaction time of 1 minute? This is an extremely short time which makes very difficult to assess stability or the kinetics of the system. Authors have to increase the initial amount of NH₃, as now is limited to only 12 mL. This will also demonstrate the applicability of the system.

R: We sincerely thank the reviewer for raising this important issue. The choice of a 1-minute total reaction time was a carefully considered decision, primarily aimed at ensuring that we could accurately and reliably measure the intrinsic reaction rate of the system, while minimizing the effects of product concentration changes and revealing its true kinetics.

This short duration is critical because it ensures that the reaction occurs in the low-conversion region (<20%). Under these conditions, the effects of reactant (NH₃) consumption and product (H₂, N₂) accumulation on the reaction rate are minimized, allowing the rate to remain constant and unaffected by mass transfer limitations or shifts in thermodynamic equilibrium. As confirmed by our supplementary data (Figure R12), when the reaction time exceeds 1 minute, the apparent reaction rate begins to decline due to the increasing consumption of reactants and product inhibition effects. At this point, it becomes difficult to distinguish whether the observed decline is due to the intrinsic activity of the catalyst or concentration-induced kinetic decay. Therefore, the selection of 1 minute as the experimental time was a thoughtful choice to capture the most accurate and representative kinetic data, free from significant concentration changes that could introduce confusion.

Additionally, we fully understand and agree with the reviewer's point that a 1-minute short-term test is insufficient to assess the long-term stability of the catalyst, which is crucial for evaluating its practical application potential. To address this concern and more comprehensively demonstrate the catalyst's performance, we have followed the reviewer's suggestion and added a flow reactor cycling experiment lasting over 50 hours in response to question (d).

Figure R12. Ammonia conversion over time on Ni-CeO_{2-x}/CNTs.

f) Most of the works on photo-thermal ammonia decomposition have been performed under continuous-flow configuration, which is closer to real application. Authors should perform their activity tests under these conditions to contextualize their results.

R: We sincerely thank the reviewer for this valuable suggestion. As outlined in our previous response, we have fully

incorporated the experiments conducted in the continuous-flow reactor in the revised manuscript. Additionally, we have compared the ammonia decomposition activity and long-term cycling stability of the catalysts developed in this study with those of classic catalysts reported in the literature under identical continuous-flow conditions. Once again, we greatly appreciate the reviewer's constructive feedback, which has significantly enhanced the completeness and rigor of our research.

g) In line 173, authors claim that the conversion surpasses the thermodynamic equilibrium. As we cannot trick thermodynamics, this means that the actual temperature of the active sites is higher than that registered by the thermocouple (155 degrees Celsius). Authors claim that the temperature was also monitored by IR but there aren't any IR thermal images neither in the main manuscript nor in the SI. I suggest authors to back-calculate the actual temperature of the active sites in their system, for instance using their actual conversion at the equilibrium. See the work from prof. Zhang (doi.org/10.1002/anie.202304452)

R: We sincerely thank the reviewer for this important suggestion. In photothermal catalysis, both light and thermal effects can operate separately or synergistically. As a result, photothermal catalysis can be classified into different types based on their specific reaction pathways, namely thermochemical or photochemical pathways. Our work falls under **thermal-assisted photocatalysis**, where the primary reaction pathway is photochemical. In this reaction, light excites the catalyst into an electron-excited state, and thermal energy plays a supporting role by facilitating the migration of photogenerated charge carriers and enhancing the mass transfer rate, thereby accelerating the reaction process.

Unlike **photo-assisted thermal catalytic reactions**, where light is primarily used to generate heat (similar to traditional thermal catalysis) with minimal contribution from the photochemical pathway, the main driving force in our system is the energy of photons (photocatalysis-dominated). Photocatalytic reactions harness photon energy to drive "uphill" reactions ($\Delta G > 0$), enabling reactions that would typically require high temperatures to occur at low temperatures or even at room temperature. Many studies have demonstrated that photocatalysis can break the limitations of traditional thermal catalysis, achieving conversion rates that surpass the thermodynamic equilibrium limits at the same temperature (doi.org/10.1016/j.nanoen.2024.109401, doi.org/10.1038/s41929-019-0419-z, doi.org/10.1002/anie.202108870, doi.org/10.1038/s41929-019-0419-z).

In photocatalysis, the input of light energy drives the system into a non-thermal equilibrium state, allowing significant conversion rates at apparent temperatures much lower than those required for thermal catalysis. This phenomenon—where the apparent conversion rate exceeds the thermodynamic equilibrium limit at the same temperature—is one of the key features that distinguishes photocatalytic reactions from traditional thermal catalysis.

We fully understand the reviewer's concern that an apparent conversion rate surpassing the thermodynamic equilibrium limit at the given temperature could indicate that the local temperature of the active sites is higher than the average bulk temperature. To ensure the absolute rigor of our conclusions and account for potential temperature measurement errors, we have removed statements and related figure (**Fig. 2B**) regarding "conversion surpassing thermodynamic equilibrium" from the revised manuscript.

Once again, we deeply appreciate your invaluable suggestion, which has greatly helped us improve the rigor and scientific accuracy of our manuscript.

Responses to the Reviewers' Comments and the Corresponding Revisions

Referee 1

A minor comment: while most figures are high-resolution vector graphics, the atomic configuration panels, particularly Fig. 4E, make it difficult to distinguish among intermediates. I suggest that the authors enhance the resolution of these images.

Aside from this, the authors have adequately addressed all of my comments, and I am pleased to recommend the manuscript for publication in Nature Communications.

R: We thank the reviewer for their positive recommendation and for this helpful suggestion. In response, we have revised Fig. 4E (and all similar atomic configuration panels) at a higher resolution. The local views of the optimized structures in Fig. 4E have enlarged to help readers clearly distinguish the intermediates. The complete structural diagrams are provided in Supplementary Fig. 26.

Revised Fig. 4E

Referee 3

I appreciate the time and effort of the authors to address my comments. Overall, I think the manuscript improved its clarity after this revision.

Regarding my question about reaction time, now it is clear that authors used 1 min reaction time to measure initial reaction rates. And the actual conversion of those experiments is 91 % after 20 minutes of reaction (as per new figure R12). Still it is quite surprising that the catalyst is able to achieve a 92 % conversion under continuous-flow (24 mL/min flow rate of pure NH₃) only at 300 degrees Celsius. Specially taking into account that under batch conditions a total reaction time of 20 min was necessary to convert 91 % of 12 mL of NH₃.

Authors declined to perform back-calculation of reaction temperature using equilibrium conversions, but probably these data could be useful to estimate the real temperature, which most likely will be higher than 300 degrees Celsius.

R: We sincerely thank the reviewer for their thoughtful comments and the opportunity to further clarify the apparent discrepancy in conversion between the batch and flow systems. The high conversion achieved in the continuous-flow reactor at 300 °C is reliable and can be well explained by several key experimental differences.

First, the batch reaction yielding 91% conversion in 20 minutes was conducted under a light intensity of 1.4 W/cm², while the continuous-flow experiment was performed under a significantly higher intensity of 3.5 W/cm², which raised the catalyst temperature from 155 °C to 306 °C. This increase in photon flux greatly enhances the photocatalytic driving force. Moreover, the batch system is inherently limited by dead volume and inefficient contact between the catalyst and reactant, which slows reaction kinetics. In contrast, the continuous-flow configuration ensures thorough and continuous contact between NH₃ and the catalyst, fully leveraging its high intrinsic activity and enabling more efficient conversion. Therefore, the 92% conversion obtained under flow—supported by both higher photon flux and superior mass transfer—is well justified.

In addition, we fully understand the reviewer's concern regarding the actual reaction temperature. To address this directly, we performed infrared thermal image under 3.5 W/cm² irradiation, which confirmed that the catalyst bed temperature remained consistently at 306 °C (Figure R1).

Figure R1 Infrared thermal image of Ni-CeO_{2-x}/CNT surface under 3.5 W/cm² illumination.

Furthermore, following the reviewer's suggestion, we performed a back-calculation based on the equilibrium conversion. As shown in Figure R2, the thermodynamic limiting temperature corresponding to 92.9% conversion is approximately 231 °C (blue point). However, the measured temperature in our continuous-flow system was 306 °C (orange point), and the conversion obtained under this condition is slightly below the theoretical maximum conversion at this temperature. This result confirms that the reported macroscopic temperature is consistent and credible.

Figure R2 The limitation of the thermal system for ammonia decomposition (blue line).

In conclusion, the high performance in the continuous-flow reactor arises from the combined effect of enhanced photon flux (3.5 W/cm^2) and optimized reactant-catalyst contact in the continuous-flow configuration. We have also added relevant explanations in the revised manuscript regarding the photo-thermal temperature and its relation to the thermodynamic limit: “Under 3.5 W cm^{-2} irradiation, a photothermal temperature of 306 °C and a maximum conversion of 92.9% were achieved, which corresponds to a thermodynamic equilibrium temperature of 231 °C ”.